# Involvement of Cardiorespiratory Capacity on the Acute Effects of Caffeine on Autonomic Recovery

**DOI:** 10.3390/medicina55050196

**Published:** 2019-05-23

**Authors:** Luana A. Gonzaga, Luiz C. M. Vanderlei, Rayana L. Gomes, David M. Garner, Vitor E. Valenti

**Affiliations:** 1Postgraduate Program in Physiotherapy, Universidade Estadual Paulista, Presidente Prudente 19060–900, Brazil; lcm.vanderlei@unesp.br (L.C.M.V.); rayana.loch@hotmail.com (R.L.G.); david.garner1@gmail.com (D.M.G.); vitor.valenti@gmail.com (V.E.V.); 2Cardiorespiratory Research Group, Department of Biological and Medical Sciences, Faculty of Health and Life Sciences, Oxford Brookes University, Headington Campus, Gipsy Lane, Oxford OX3 0BP, UK

**Keywords:** caffeine, autonomic nervous system, exercise, recovery of physiological function, physical fitness

## Abstract

*Background and objectives*: As a result of ergogenic properties, caffeine has been increasingly taken prior to physical exercise, yet its effects on post-exercise recovery, considering the differences in the cardiorespiratory capacity of the individuals, has not yet been studied or fully elucidated. Optimizing the post-exercise recovery can convey advantages to physical activity practitioners. We evaluated the acute effects of caffeine on heart rate (HR) autonomic control recovery following moderate aerobic exercise in males with different cardiorespiratory capacities. *Materials and Methods*: We split young adult men into two groups based on their various oxygen consumption peaks (VO_2_ peak): (1) Higher VO_2_ (HO): Sixteen volunteers, peak VO_2_ > 42.46 mL/kg/min and (2) Low VO_2_ (LO): Sixteen individuals, VO_2_ < 42.46 mL/kg/min). The volunteers were submitted to placebo and caffeine protocols, which entailed 300 mg of caffeine or placebo (starch) in capsules, followed by 15 min of rest, 30 min of moderate exercise on a treadmill at 60% of the VO_2_ peak, followed by 60 min of supine recovery. Heart rate variability (HRV) indexes in the time and frequency domains were examined. *Results*: Effect of time for RMSSD (square root of the average of the square of the differences between normal adjacent RR intervals) and SDNN (standard deviation of all normal RR intervals recorded in a time interval) was achieved (*p* < 0.001). Significant adjustments were observed (rest versus recovery) at the 0 to 5th min of recovery from exercise for the LO during the placebo protocol and at the 5th at 10th min of recovery for the caffeine protocol. For the HO in both procedures we found significant alterations only at the 0 to 5th min of recovery. *Conclusion:* Caffeine delayed parasympathetic recovery from exercise in individuals with lower cardiorespiratory capacity.

## 1. Introduction

Recovery from exercise is a critical period to investigate the possibility of cardiovascular events [1,2], including abnormal blood pressure responses, heart rhythm disorders and abnormal or irregular heartbeats [3,4].

The evaluation of the autonomic regulation of one’s heart rate (HR) is a worldwide used procedure to detect the possibility of cardiovascular complications [2]. This is because the autonomic nervous system (ANS) is responsible for regulating a large part of the body’s’ visceral functions and performs an important role in the control of physiological responses to stresses, including physical exercise [1].

With this in mind, HR variability (HRV) is a well-recognized method that assesses cardiac autonomic function. It describes the oscillations of consecutive heartbeat intervals (RR intervals), that are related to the ANS influences on the sinus node. HRV is a non-invasive measurement that indicates the ability of the heart to respond to different physiological and environmental stimuli, in addition to compensating for changes because of different diseases [5,6,7].

The ANS is similarly influenced by the consumption of stimulants such as caffeine, since caffeine blocks adenosine receptors (A1 and A2) and activates the sympathetic ANS; thus, it is an important stimulator of the ANS. This mechanism promotes increased release of catecholamines from the adrenal medulla [6], stimulating glycogenolysis and lipolysis [6,7].

On account of caffeine’s ergogenic properties (enhancing physical performance), caffeine has been progressively used as a supplement prior to physical training by athletes and physically active individuals in order to promote better performance [8]. Some studies have reported the use of caffeine to extend the time to exhaustion [7,9], muscle strength, metabolism [7] and improvement in cognitive performance, such as increased vigilance, attention and the reduction of reaction times [10].

On account of its ergogenic (enhancing physical performance) properties, caffeine has been progressively used as a supplement prior to physical training by athletes and physically active individuals in order to promote better performance [9,10]. Also, caffeine appears to promote influences on exercise-induced responses in sedentary individuals. Wallman et al. [11] and Laurence et al. [12] observed that caffeine consumption promoted positive effects on oxygen consumption and on energy expenditure during exercise on a stationary bicycle.

Caffeine ingestion results in tachycardia [13]. Though seemingly innocuous, these alterations were believed to be related to elevated risks of arrhythmias [14]. A study by Magkos et al. [14] emphasized that the physiological changes triggered by the combination of caffeine and exercise may be unfavorable without familiarity of the cardiovascular parameters and the individuals’ characteristics. These findings suggest that depending on physical activity intensity, caffeine may become a contributing factor for harmful cardiovascular complications following exercise in subjects with cardiovascular diseases [15].

As recognized above, caffeine supplementation in different populations is one of the ergogenic strategies investigated in the scientific research literature [3,7,11,15]. Considering that caffeine is consumed widely [16], detailed information regarding its impact on health is necessarily investigated, especially for its use in exercise. Still, the involvement of cardiorespiratory capacity in the effects of caffeine on autonomic recovery following exercise have not been fully elucidated.

Optimizing the post-exercise recovery can convey advantages to physical activity practitioners. Studies that evaluate this process can provide important information that allow the enhancement of training and rehabilitation protocols and the prevention of deleterious cardiac events.

In this context, the present study intends to evaluate the acute effects of caffeine intake before exercise on the recovery of autonomic HR control following moderate aerobic exercise in males with different cardiorespiratory capacities. We split young adult men into two groups based on their various oxygen consumption peaks (VO_2_ peak).

## 2. Materials and Methods

This is a prospective, crossover, single-blind, controlled and randomized trial performed in 40 male volunteers (23.59 ± 3.45 years), recruited through social media. Smokers, alcoholics, people with cardiopulmonary, neurological, musculoskeletal disorders or other pathological conditions that prevented the realization of protocols were not included. Only series of consecutive heart beats (RR intervals) with more than 95% of sinus beats were included in the study.

The volunteers were informed about the objectives and procedures, they signed a confidential and informed consent form. The experimental procedures were approved by the Ethics Committee in Research of the Paulista State University (Protocol number: CEP-2200/11) and followed the 466/12 resolution (National Health Council (12/12/2012). The present study is registered in the Clinical Trials network (identification code NCT02917889 on 19 September 2016).

### 2.1. Study Design

The study procedures were divided into three protocols with a minimum interval of 48 h between them to allow the physical recovery of the participants. In the first stage we obtained body weight (digital scale, Welmy W 200/5, Brazil) and height (stadiometer ES 2020, Sanny, Brazil).

The protocols were performed between 17:30 and 21:30 to standardize the circadian influences. The temperature was between 23 °C and 24 °C, ambient humidity between 60% and 70%. Before the experimental procedure, all volunteers were advised to not consume alcoholic and caffeinated beverages and to not practice intense exercise at least 24 h before each protocol, and were recommended to not consume a heavy meal before the data collection.

The first study protocol was the maximal aerobic power (VO_2_ peak) test to determine the exercise intensity in following stages and division of the groups. The other phases were the placebo and caffeine protocols, whose order of execution was established through a randomization process using a coin. The volunteers were blinded during their protocol and were not informed about the order of these protocols; However, the researcher was not blinded at any time in the study regarding the order of protocols and capsules. The volunteers were divided into two groups based on the median value of VO_2_ peak): (1) Higher VO_2_ peak (HO) composed of volunteers with VO_2_ peak > 42.46 mL/kg/min, and (2) Lower VO_2_ peak (LO) composed of volunteers with VO_2_ peak <42.46 mL/kg/min. This division was based on the optimal VO_2_ peak considered for values >43 [17].

### 2.2. Maximal Aerobic Power Test

So as to establish the division of the groups and the intensity of the exercise to be prescribed in the exercise protocols, the cardiopulmonary exercise test on treadmill (Inbramed, MASTER CI, Brazil) was performed using the Bruce incremental protocol [18]. The analysis of expired gases was completed using the Quark PFT commercial system (Comend, Rome, Italy), obtaining the VO_2_ peak established as the highest VO_2_ peak achieved during the test.

### 2.3. Caffeine and Placebo Protocol

Prior to initiating the protocols, the HR monitor (Polar RS800CX, Finland) was strapped to the volunteers’ chest to register HR beat-to-beat, followed by an intake of 300 mg of caffeine or 300 mg of starch (placebo) in indistinguishable capsules, in accordance with the protocol selected. We decided to prescribe 300 mg of caffeine because this concentration is considered safe. The Food and Drug Administration (FDA) [19] indicated that 300 mg is within the maximum allowed per day and the 2015–2020 Dietary Guidelines for Americans state up to 400 mg/day is safe [19].

After ingesting the caffeine or placebo capsules, subjects performed an initial rest in supine position for 15 min. Next, the volunteers performed exercise on a treadmill at five km/h and 1% of slope in the first five minutes for warm up and a consecutive 25 min of exercise at 60% of the HR achieved at the VO_2_ peak with an inclination of 1%. After completing the exercise protocol, the volunteers remained at rest in the supine position under spontaneous breathing and were monitored for 60 min.

### 2.4. Heart Rate Variability Analysis

The analysis of HRV was accomplished in the whole experimental protocol through a heart rate monitor (Polar RS800CX, Finland), previously validated equipment to register heart rate beat to beat [20]. HRV indexes were determined at the following periods: Tenth to 15th minute of rest and from exercise recovery (Rec): Rec1 (0 to 5 min), Rec2 (5 to 10 min), Rec3 (15 to 20 min), Rec4 (25 to 30 min), Rec5 (35 to 40 min), Rec6 (45 to 50 min) and Rec7 (55 to 60 min). The volunteers were instructed to stay awake and remain silent on spontaneous breathing in the supine position.

We performed digital filtering of RR intervals complemented by manual filtering to eliminate premature ectopic beats and artifacts, and only series with more than 95% of sinus beats were accepted in the study. For HRV analysis exactly 256 stable consecutive RR intervals were considered. To analyze HRV we applied linear methods in the time and frequency domain. The time domain indexes included were: RMSSD (square root of the average of the square of the differences between normal adjacent RR intervals) and SDNN (standard deviation of the average of all normal RR intervals). The Poincaré plot indexes: SD1 (standard deviation of the instantaneous rate variability the rhythm) and SD2 (long-term standard deviation of RR intervals) [5] were computed.

We analyzed the frequency domain based on the low (LF, 0.04 to 0.15 Hz) and high frequency (HF-0.15 to 0.40 Hz) spectral indexes in normalized units (n.u.) calculated through the Fast Fourier Transform (FFT), and the ratio between the low and high frequency (LF/HF). The Kubios software (Kunios HRV^®^, Biosignal Analysis and Medical Image Group, Department of Physics, University of Kuopio, Finland) [21] was used to analyze the linear indexes in the time and frequency domains and the Poincaré plot [5].

### 2.5. Data Analysis

The sample size was calculated through a pilot test based on the RMSSD HRV index and the online software from the website www.lee.dante.br was required. The significant difference in magnitude was assumed for 14.11 ms, with a standard deviation of 12.8 ms, an alpha risk of 5% and a beta of 80%. The sample size calculation provided a minimum of 13 individuals per group. For the cataloguing, the population descriptive statistics were calculated and the results were presented as mean, standard deviation, median, minimum and maximum values.

To compare variables between groups, data normality was determined by the Shapiro–Wilk test, and when the normal distribution was acknowledged, Student’s *t*-test for unpaired data was applied. In circumstances where the normal distribution was not recognized the Mann–Whitney test was applied. Differences in these tests were considered statistically significant when the *p*-value was less than 0.05 (<5%). The effect size was calculated through Cohen’s d to verify the magnitude of the difference between variables. A large effect size was considered for values above 0.8, mean values were between 0.79 and 0.5, and small were for values less than 0.2 [22].

Comparisons of HRV indexes between protocols (caffeine versus placebo) and moments were made by analysis of variance techniques to model repeated measures on a two factors scheme (two-way ANOVA). The repeated measures data were checked for sphericity violation using Mauchly’s test and the Greenhouse–Geisser correction was directed when the sphericity was violated. To analyze the moments (rest vs. exercise vs. recovery periods) we applied ANOVA for repeated measurements followed by the Bonferroni post-hoc for parametric distribution or the Friedman test followed by Dunn’s post-test for non-parametric distribution. Statistical significance was set at *p* < 0.05 or (<5%) for all tests.

## 3. Results

Table 1 displays the anthropometric characteristics and the values obtained in the maximal aerobic power test of the groups (Figure 1) with a higher VO_2_ peak and with a lower VO_2_ peak. There were no significant differences in relation to the anthropometric characteristics of the groups except in relation to the VO_2_ peak, as expected since this variable was considered in the division of the groups.

In relation to the response of frequency domain HRV indexes we observed a time effect (*p* < 0.0001) between Rec1 and rest for the LO group in both protocols and for HO only in the placebo protocol, yet no effects were observed in protocol interaction for LF n.u. (*p* = 0.798) and HF n.u. (*p* = 0.727) indexes, and we detected protocol interaction only for the LF/HF ratio (*p* = 0.031). No effect among protocols were observed for the frequency domain indexes: LF n.u. (*p* = 0.163), HF n.u. (*p* = 0.165) and LF/HF (*p* = 0.094). The response of frequency domain HRV indexes are shown in Figure 2.

In relation to the time domain HRV analysis, to SDNN index, a time effect (*p* < 0.0001) was found, yet no significant differences were achieved in protocol interaction (*p* = 0.601) and there was no effect between protocols (*p* = 0.93).

Regarding RMSSD and SD1 indexes, no effects in protocol interaction (RMSSD: *p* = 0.524; SD1: *p* = 0.359) and no effect among protocols (RMSSD: *p* = 0.657; SD1: *p* = 0.786) were observed. Still, significant differences were achieved between rest and Rec1 in the placebo protocol and between Rec1 and Rec2 in the caffeine protocol for the LO group. Regarding the HO group, in the placebo and caffeine protocols, significant differences were attained only between rest and Rec1. For the SD2 index we found a time effect (*p* < 0.0001) with a significant difference observed between rest and Rec6 for the LO group in the placebo protocol. No significant changes were detected in the protocol interaction (*p* = 0.663) and amongst the protocols (*p* = 0.496). Figure 3 illustrates the time domain HRV analysis at rest and during recovery from exercise.

## 4. Discussion

As a main finding, we found evidence that the caffeine effects on HRV during recovery from exercise were more intense in subjects with a lower VO_2_ max. The analysis of RMSSD and SD1 indexes in the LO group during the caffeine protocol disclosed that the vagal HR control recovery was slower, occurring only in Rec3 (15th to 20th minute after exercise). While in the placebo protocol the parasympathetic HR regulation recovery was observed in Rec2 (5th to 10th minute after exercise). Regarding the HO group, recovery was observed in Rec2 in both protocols.

During exercise, HRV reduction is a result of lessened parasympathetic modulation and increased sympathetic stimulation [23]. During recovery from exercise HRV may be influenced by several factors, including intensity, duration, exercise modality [24] and cardiorespiratory capacity [25]. Individuals with higher physical fitness have the most rapid post-exercise recovery capacity [26], which could be attributed to higher enzyme concentrations, quantity of mitochondria, myoglobin and increased capillary density in the muscle fibers supporting a greater blood flow. These factors may contribute to an increase in the VO_2_ during exercise and a lower lactic acid buildup [27].

Regarding the ANS, individuals with a higher cardiorespiratory capacity have a faster parasympathetic re-entry and a smaller sympathetic activity at rest [25], which contributes to a more rapid post-exercise recovery [25,28]. These adaptations can be attributed to the chronic effects of physical exercise, such as decreased norepinephrine secretion and increased arterial baroreflex sensitivity [29,30].

In this study, the group comprised of subjects with lower VO_2_ peak presented with a slower reactivation of parasympathetic activity in the caffeine protocol, signifying that caffeine delayed the vagal reactivation in individuals with lower cardiorespiratory capacity, which can be accredited to their lower physical conditioning.

Regarding the frequency indexes, in the LO group in both protocols and in the HO group in the placebo protocol, differences between at rest and 0 to 5 min of recovery from exercise were observed in comparison to rest for all indexes. We observed a decrease in the HF n.u. and an increase of LF n.u. between rest and 0 to 5 min after exercise, demonstrating a physiological behavior highlighted by a decrease in the parasympathetic modulation immediately after exercise (Rec1) with progressive return to the basal values during recovery. Correspondingly, the sympathetic modulation LF n.u. was elevated immediately after exercise (Rec1) and returned to baseline values during recovery.

According to our results, there was no modification between at rest and recovery when evaluating frequency-domain HRV indexes in the caffeine protocol in the HO group. This implies that caffeine was unable to influence the response of these indexes in individuals with higher cardiorespiratory fitness.

A few studies had investigated the effects of caffeine on post-exercise recovery. Bunsawat et al. [3] observed that after consumption of caffeine (400 mg capsules), the QT interval of healthy young men remained prolonged during the recovery after a maximal exercise, when compared to the placebo protocol, signifying a greater stimulation of sympathetic modulation promoted by caffeine. Yeragani et al. [31] reported a decrease in the power of the HF band during exhaustive exercise on a cycle-ergometer after caffeine consumption, which has been given an exaggerated vagal withdrawal promoted by caffeine [31].

Recently, we have revealed that caffeine affected HRV in exercise [32]. It was observed that an intake of caffeine (300 mg) before exercise was able to slow HRV recovery following moderate (60% of VO_2_ peak) aerobic exercise on a treadmill. These results indicated that caffeine in capsules influenced the vagal regulation of HRV.

It is vital to emphasize that these studies did not take into account the cardiorespiratory capacity of volunteers and enforced high intensity exercises in the protocols. In this way, the moderate-intensity exercise originated in this study may not have promoted sufficient physical stress to originate more sensitive changes in the recovery of autonomic modulation following exercise.

Similar results were detected by Thomas et al. [33], who investigated the effects of caffeine intake (300 mg capsules) on the frequency-domain HRV indexes during recovery from submaximal exercise, considering the C*yp1a2* gene polymorphism that is responsible for the rate of caffeine metabolization. The study stated that RMSSD recovery rate was slower in individuals who had the gene polymorphism. The authors were unable to find significant modifications between the groups when studying the frequency domain indexes HF n.u., LF n.u. and LF/HF ratio. Concerning global HRV, no significant differences were observed among protocols or for the protocol interaction, suggesting that exercise intensity and caffeine were insufficient to cause significant deviations in global HRV.

Our study highlights some important methodological points. The sample was comprised of healthy young men in order to avoid the influence of sexual hormones. For this reason, our results cannot be applied to females owing to variations in caffeine metabolism during the menstrual cycle [32] or to individuals with cardiac diseases who are prescribed medications that affect the ANS. Yet the selection of the sample performed using rigorous exclusion criteria supports our results. Although it should be noted that this is a single-blind study, and we failed to perform a double-blind design.

Our investigation contributes new material on mechanisms related to the impact of caffeine on post-exercise recovery in diverse populations. Specifically, in this case, healthy young men with lower VO_2_ peak were more susceptible to being affected by caffeine during recovery from moderate aerobic exercise. This investigation focused attention to the measurement of VO_2_ peak, a method that provides relevant information for the cardiorespiratory capacity. We did not evaluate plasma catecholamine concentrations or the sympathetic nerve activity; nevertheless, we imposed HRV, a simple, reliable, non-invasive method and one of the most promising quantitative markers of autonomic heart rate balance [34].

## 5. Conclusions

Our findings reported that caffeine delays autonomic recovery from moderate aerobic exercise in men with lower cardiorespiratory capacity. Taken together, men with lower physical fitness should be cautious when ingesting caffeine before exercise.

## Figures and Tables

**Figure 1 medicina-55-00196-f001:**
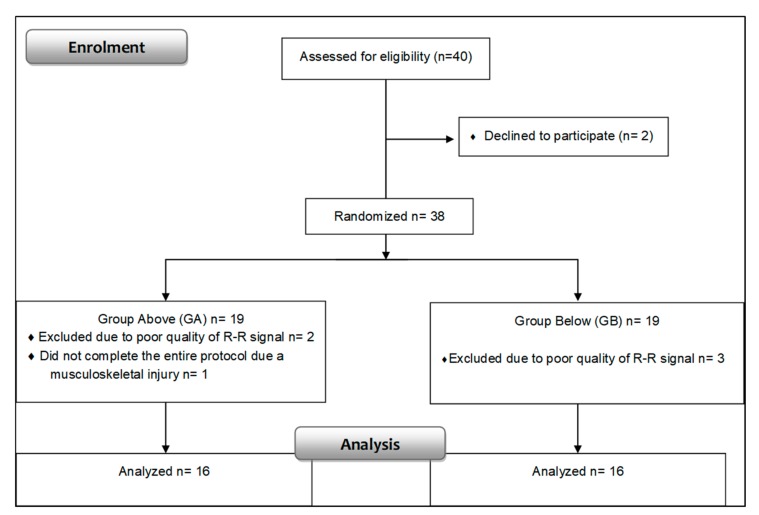
Displays the flow diagram signifying the progress of all participants through the trial.

**Figure 2 medicina-55-00196-f002:**
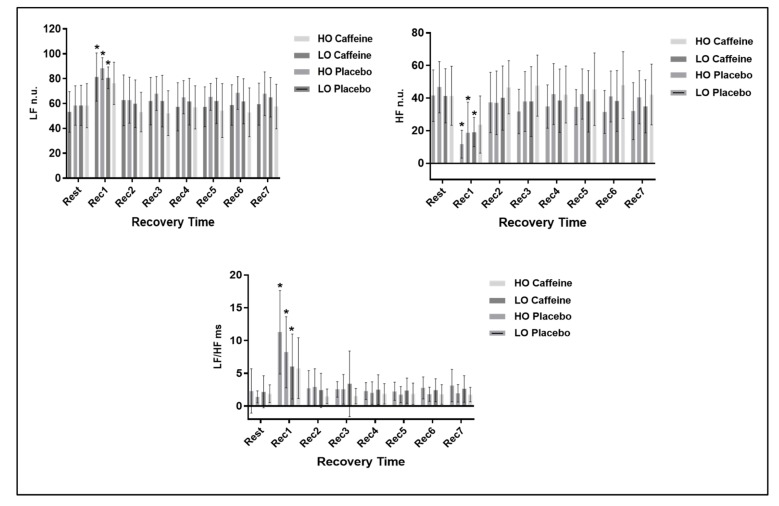
Displays the the response of frequency domain HR variability (HRV) indexes at rest and during recovery from exercise.

**Figure 3 medicina-55-00196-f003:**
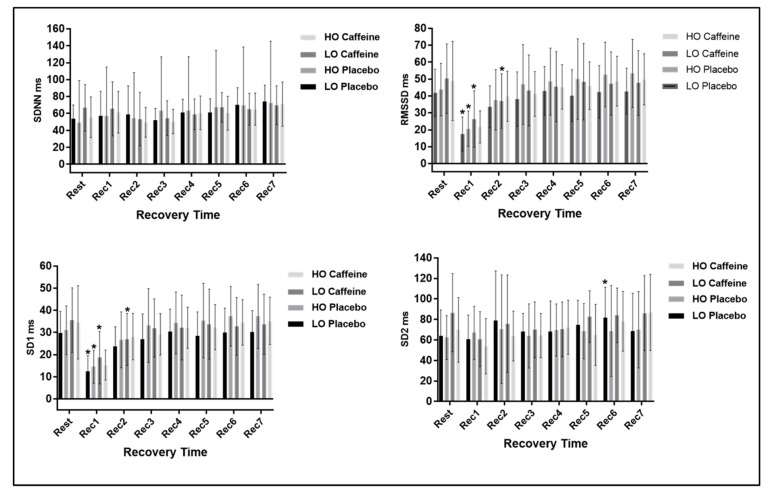
Displays the response of time domain HRV indexes at rest and during recovery from exercise.

**Table 1 medicina-55-00196-t001:** Values (mean, minimum and maximum) and standard deviations of the anthropometric variables and the cardiopulmonary exercise testing.

	Mean ± SD	Min–Max	Mean ± SD	Min–Max	p	C	Effect Size
**Age (years)**	23.69 ± 3.75	[19–30]	23.50 ± 3.25	[19–30]	0.801	0	Small
**Height (m)**	1.80 ± 0.04	[1.71–1.85]	1.80 ± 0.09	[1.59–1.96]	0.866	0	Small
**Mass (kg)**	78.38 ± 6.92	[62–91]	79.38 ± 16.01	[56–100]	0.820	0.08	Small
**BMI (kg/m²)**	24.33 ± 2.04	[19.35–29.04]	24.48 ± 3.50	[19.94–27.70]	0.844	0	Small
**VO_2_ peak (mL/kg/min)**	53.32 ± 8.79	[42.68–72.79]	34.69 ± 6.92	[23.12–42.16]	**0.000 ***	2.6	Large
**Peak HR (bpm)**	186.44 ± 9.71	[160–199]	185.50 ± 13.42	[154–211]	0.822	0.08	Small
**60% peak HR (bpm)**	111.94 ± 5.12	[100–119]	111.06 ± 8.01	[92–126]	0.715	0	Small

Legend: HO = group with higher VO_2_ peak; LO = group with lower VO_2_ peak; SD = standard deviation; C: Cohen’s; BMI = body mass index; kg = kilogram; m = meter; VO_2_ peak = peak oxygen consumption; ml = milliliter; min = minutes; HR = heart rate; bpm = beats per minute. * Statistical significance between groups (Mann-Whitney test).

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
