# Peer review of "Involvement of Cardiorespiratory Capacity on the Acute Effects of Caffeine on Autonomic Recovery"

_medicina, 2019, doi:10.3390/medicina55050196_

Round 1
Reviewer 1 Report
The title can be shortened: Not needed is "the ïnvolvement of oxygen consumption" because the main focus is on the post exercise recovery.
I should use maximal aerobic power in stead of maximal aerobic capacity because peak oxygen uptak per minute is a power function.
line 255 in conclusion: "to be cautious must be "changed in "be cautious"
Author Response
Dear Dr.,
We appreciate your important and careful revision. We have revised the manuscript to improve it based on your comments. The added or modified words, phrases, and sentences are in red.
Comments and Suggestions for Authors:
· “The title can be shortened: Not needed is "the ïnvolvement of oxygen consumption" because the main focus is on the post exercise recovery”.
Answer: We thank the recommendation, considering that the cardiorespiratory capacity was the novelty of our study, we remade the title to:
“Involvement of cardiorespiratory capacity on the acute effects of caffeine on autonomic recovery”.
· “I should use maximal aerobic power in stead of maximal aerobic capacity because peak oxygen uptak per minute is a power function”.
Answer: We agreed and appreciate the recommendation. Changes are highlighted in red.
· “line 255 in conclusion: "to be cautious must be "changed in "be cautious".
Answer: Thank you for the correction. Changes are highlighted in red.
Reviewer 2 Report
There are no information how the group was divided (into research and control groups); who coded capsules (coffeine / placebo), who and when decoded it.
In Results chapter Authors describes tables and figures, however text should to the description of the results of research.
Figures (2 and 3) are difficult to rear and analysis.
There are lack of many important articles (especially regarding post-exercise recovery) in references.
Article prepared regardless, eg. there are no titles of figures.
Author Response
Dear Dr.,
We appreciate your important and careful revision. We have revised the manuscript to improve it based on your comments. The added or modified words, phrases, and sentences are in red.
Comments and Suggestions for Authors
· “There are no information how the group was divided (into research and control groups); who coded capsules (coffeine / placebo), who and when decoded it”.
Answer: We added a paragraph that described the group division.
Line 104-108: “The other phases were the placebo and caffeine protocols, whose order of execution was established through a randomization process using a coin. The volunteers were blinded during their protocol and were not informed about the order of these protocols; However, the researcher was not blinded at any time in the study regarding the order of protocols and capsules”.
· “In Results chapter Authors describes tables and figures, however text should to the description of the results of research”.
Answer: We agreed and appreciate the recommendation. We made some changes to the results to improve the description.
· “Figures (2 and 3) are difficult to rear and analysis”.
Answer: We remade figures resolution. We hope this version can be accepted
.
· “There are lack of many important articles (especially regarding post-exercise recovery) in references.”
Answer: We appreciate the recommendation. We have added some studies in your article, as changes are highlighted in red
Line 55-60: “On account of its ergogenic (enhancing physical performance) properties, caffeine has been progressively used as a supplement prior to physical training by athletes and physically active individuals in order to promote better performance [10,11]. Also, caffeine appears to promote influences on exercise-induced responses in sedentary individuals. Wallman et al. [12] and Laurence et al. [11] observed that caffeine consumption promoted positive effects on oxygen consumption (VO2) and on energy expenditure during exercise on a stationary bicycle.”
Line 258-261: “Recently, we have revealed that caffeine affected HRV in exercise [28]. It was observed that an intake of caffeine (300 mg) before exercise was able to slow HRV recovery following moderate (60% of VO2 peak) aerobic exercise on a treadmill. These results indicated that caffeine in capsules influenced the vagal regulation of HRV.”
· “Article prepared regardless, eg. there are no titles of figures”.
Answer: Thank you for the correction. Changes are highlighted in red.
Reviewer 3 Report
The purpose of this investigation was to examine the influence of acute caffeine supplementation on heart rate recovery following moderate-intensity exercise in two groups of males with different levels of aerobic capacity. The authors reported that the caffeine supplement delayed parasympathetic recovery following exercise in the lowered aerobic capacity group.
In general, the manuscript has some lacks and I have major concerns related to the overall rationale and methodology.
Specific concerns:
1. Adding a purpose statement to the abstract would be helpful.
2. The abstract in general appears to be sloppy. For example, the “2” in VO2 is not subscripted, “kg” and “Kg” are both used to describe ml/kg/min, and abbreviations are used that are not clearly defined. When authors do not appear to pay attention to these finer details, it makes the reviewer wonder what other parts of the study (e.g. data collection) were not handled meticulously.
3. The same comment above can be applied to the Introduction section. More care needs to be considered when proofreading.
4. The Introduction is choppy and the logic does not flow from one paragraph to the next. Also, the rationale for caffeine’s influence on autonomic recovery is not described in the Introduction. This is a vital piece to make the manuscript compelling. Also, how is cardiorespiratory/aerobic capacity related to this? The rationale for using two different groups is not explained.
5. Why was the study single-blind? Is it possible that not having a double-blinded study negatively influence the results?
6. The two groups were divided based on median VO2peak, but is the median value of 42.5 ml/kg/min physiologically relevant to the research question? There is no information provided that addresses this question.
Author Response
Dear Dr.,
We appreciate your important and careful revision. We have revised the manuscript to improve it based on your comments. The added or modified words, phrases, and sentences are in red.
Specific concerns:
· “Adding a purpose statement to the abstract would be helpful”.
Answer: We appreciated the recommendation. We add in the summary the purpose of the study.
Line16-18: “Optimizing the post-exercise recovery can convey advantages to physical activity practitioners, in this context, the present study intends to evaluate the acute effects of caffeine intake before exercise on the recovery of autonomic HR control following moderate aerobic exercise in males with different cardiorespiratory capacities.”
· “The abstract in general appears to be sloppy. For example, the “2” in VO2 is not subscripted, “kg” and “Kg” are both used to describe ml/kg/min, and abbreviations are used that are not clearly defined. When authors do not appear to pay attention to these finer details, it makes the reviewer wonder what other parts of the study (e.g. data collection) were not handled meticulously”.
Answer: Corrections are in red.
· “The same comment above can be applied to the Introduction section. More care needs to be considered when proofreading”.
Answer: Corrections are in red.
· “The Introduction is choppy and the logic does not flow from one paragraph to the next. Also, the rationale for caffeine’s influence on autonomic recovery is not described in the Introduction. This is a vital piece to make the manuscript compelling. Also, how is cardiorespiratory/aerobic capacity related to this? The rationale for using two different groups is not explained”.
Answer: In order to improve Introduction section based on the rationale for caffeine’s influence on autonomic recovery, we added and remade the following paragraphs:
2nd, 3rd and 4th paragraphs:
“The evaluation of the autonomic regulation of heart rate (HR) is a worldwide used procedure to detect the possibility of cardiovascular complications [2]. This is because the autonomic nervous system (ANS) is responsible for regulating a large part of the body’s’ visceral functions and performs an important role in the control of physiological responses to stresses, including physical exercise [1].
With this in mind, HR variability (HRV) is a well-recognized method that assesses cardiac autonomic function. It describes the oscillations of consecutive heartbeat intervals (RR intervals), that are related to the ANS influences on the sinus node. HRV is a non-invasive measurement that indicates the ability of the heart to respond to different physiological and environmental stimuli, in addition to compensating for changes because of different diseases [5–7].
The ANS is similarly influenced by the consumption of stimulants such as caffeine, since caffeine blocks adenosine receptors (A1 and A2) and activates the sympathetic ANS; thus, it is an important stimulator of the ANS. This mechanism promotes increased release of catecholamines from the adrenal medulla [10], stimulating glycogen-lysis and lipolysis [6,7].”
5th and 6th paragraphs:
“On account of its ergogenic (enhancing physical performance) properties, caffeine has been progressively used as a supplement prior to physical training by athletes and physically active individuals in order to promote better performance [10,11]. Also, caffeine appears to promote influences on exercise-induced responses in sedentary individuals. Wallman et al. [12] and Laurence et al. [11] observed that caffeine consumption promoted positive effects on oxygen consumption and on energy expenditure during exercise on a stationary bicycle.
Caffeine ingestion results in tachycardia [12]. Though seemingly innocuous, these alterations were believed to be related to elevated risks of arrhythmias [13]. A study by Magkos et al., [13] emphasized that the physiological changes triggered by the combination of caffeine and exercise may be unfavorable without familiarity of cardiovascular parameters and the individuals’ characteristics. These findings suggest that depending on physical activity intensity, caffeine may become a contributing factor for harmful cardiovascular complications following exercise in subjects with cardiovascular diseases [14]. “
References added:
De Sanctis V, Soliman N, Soliman AT, Elsedfy H, Di Maio S, El Kholy M, Fiscina B. Caffeinated energy drink consumption among adolescents and potential health consequences associated with their use: a significant public health hazard. Acta Biomed. 2017;88:222-231.
Magkos F, Kavouras SA. Caffeine and ephedrine: physiological, metabolic and performance-enhancing effects. Sports Med. 2004;34(13):871-89.
Basrai M, Schweinlin A, Menzel J, Mielke H, Weikert C, Dusemund B, Putze K, Watzl B, Lampen A, Bischoff SC. Energy Drinks Induce Acute Cardiovascular and Metabolic Changes Pointing to Potential Risks for Young Adults: A Randomized Controlled Trial. J Nutr. 2019. pii: nxy303.
· “Why was the study single-blind? Is it possible that not having a double-blinded study negatively influence the results?”
Answer: The researcher knew the capsule contents, while the volunteer did know which capsule, he was ingesting. A double-blind design is considered a better method than the single-blind. We added this study limitation in Discussion section:
Line 279: “Although this is a single-blind study, we failed to perform a double-blind design.”
· “The two groups were divided based on median VO2peak, but is the median value of 42.5 ml/kg/min physiologically relevant to the research question? There is no information provided that addresses this question.”
Answer: This division was based on optimal VO2 peak considered for values >43. We added this information in Methods section, Study design, last paragraph:
“This division was based on optimal VO2 peak considered for values >43 [17].”
Reference added:
Herdy AH, Uhlendorf D. Reference values for cardiopulmonary exercise testing for sedentary and active men and women. Arq Bras
Round 2
Reviewer 3 Report
The authors have adequately addressed all of my concerns.